# LncRNA-Associated Genetic Etiologies Are Shared between Type 2 Diabetes and Cancers in the UAE Population

**DOI:** 10.3390/cancers14143313

**Published:** 2022-07-07

**Authors:** Roberta Giordo, Rida Gulsha, Sarah Kalla, George A. Calin, Leonard Lipovich

**Affiliations:** 1College of Medicine, Mohammed Bin Rashid University of Medicine and Health Sciences, Dubai 505055, United Arab Emirates; roberta.giordo@mbru.ac.ae (R.G.); rida.gulsha@students.mbru.ac.ae (R.G.); sarah.kalla@students.mbru.ac.ae (S.K.); 2Department of Translational Molecular Pathology, The University of Texas MD Anderson Cancer Center, Houston, TX 77030, USA; gcalin@mdanderson.org; 3Center for RNA Interference and Non-Coding RNAs, The University of Texas MD Anderson Cancer Center, Houston, TX 77030, USA

**Keywords:** T2D, cancer, SNP, lncRNA, UCSC Genome Browser, GWAS

## Abstract

**Simple Summary:**

The genome-wide association study (GWAS) approach to common human disease relies on single nucleotide polymorphisms (SNPs), the most common type of genetic variation in the human genome, and distinguishes “risk” and “healthy” SNP alleles. In parallel with increasing insights into the non-coding genome, emerging studies reveal that most disease-associated SNPs reside within a non-coding sequence, including lncRNA genes. These developments lay the foundation for deciphering the aetiology of complex diseases, including type 2 diabetes (T2D), and its association with an increased risk of certain cancers. Here, deploying a customized annotation pipeline on GWAS datasets, we successfully identified, and characterized, six genetic variants significantly associated with both T2D and cancer in lncRNA or genes and other non-coding regions. These variants suggest evidential proof of a shared genetic architecture between the two diseases, help to functionally explain the casual association of diabetes with cancer, and comprise a potential shortlist of candidate drug targets.

**Abstract:**

Numerous epidemiological studies place patients with T2D at a higher risk for cancer. Many risk factors, such as obesity, ageing, poor diet and low physical activity, are shared between T2D and cancer; however, the biological mechanisms linking the two diseases remain largely unknown. The advent of genome wide association studies (GWAS) revealed large numbers of genetic variants associated with both T2D and cancer. Most significant disease-associated variants reside in non-coding regions of the genome. Several studies show that single nucleotide polymorphisms (SNPs) at or near long non-coding RNA (lncRNA) genes may impact the susceptibility to T2D and cancer. Therefore, the identification of genetic variants predisposing individuals to both T2D and cancer may help explain the increased risk of cancer in T2D patients. We aim to investigate whether lncRNA genetic variants with significant diabetes and cancer associations overlap in the UAE population. We first performed an annotation-based analysis of UAE T2D GWAS, confirming the high prevalence of variants at or near non-coding RNA genes. We then explored whether these T2D SNPs in lncRNAs were relevant to cancer. We highlighted six non-coding genetic variants, jointly reaching statistical significance in T2D and cancer, implicating a shared genetic architecture between the two diseases in the UAE population.

## 1. Introduction

One of the major revelations of the Human Genome Project was that a mere 1.5% of our genome encodes proteins, while the remaining 98.5% is non-coding [1]. The HGP was succeeded by major post-genomic consortia, including ENCODE (Encyclopedia of DNA Elements) [2], which built an official GENCODE gene catalog [3], and FANTOM (Functional ANnoTation Of the Mammalian genome), which enabled the community to expand the human gene catalog with the addition of tens of thousands of validated and annotated non-coding RNA (ncRNA) genes [2,4]. The current GENCODE catalog contains 60,000 human genes, of which ~20,000 encode proteins, whereas most of the other 40,000 are comprised of several types of non-protein-coding RNA (ncRNA) genes. Long non-coding RNA (lncRNA) genes, generally defined as giving rise to RNA transcripts of more than 200 nucleotides without apparent protein-coding potential, are hence a prevalent class of human genes responsible for the most abundant and frequent type of transcripts in humans [4,5]. They have been increasingly recognized in recent years as integrally taking part in a wide range of biological and physiological processes, including transcriptional and post-transcriptional regulation of gene expression, protein translation and stability, cellular differentiation, cell lineage choice, organogenesis, and other key facets of normal development as well as disease [6]. A massive and growing amount of evidence from genomic epidemiology and population genetics has definitively indicated the association of lncRNAs with many common human diseases (e.g., cancer, diabetes, cardiovascular disease, etc.), but the underlying molecular mechanisms of this association are still poorly known [7,8]. For some lncRNAs, the connection with human diseases was made by differential gene expression analyses or functional studies as well as model organisms. A universe of other lncRNAs has been discovered from genome-wide association studies (GWAS), an approach used to statistically associate specific genetic variants with human diseases [9,10]. GWAS typically focus on associations between single-nucleotide polymorphisms (SNPs) and common diseases, but can equally be applied to any other genetic variants and any other organisms [9]. Single-nucleotide polymorphisms (SNPs) are single base-pair differences between the genomes of different individuals that occur every 500–1000 throughout the 3.3 billion bases long genome [11]. While many SNPs appear not to be associated with specific phenotypes, other SNPs’ alleles can be clearly characterized as a disease (or risk) allele and a healthy (non-risk) allele upon comparative statistical evaluation of the incidence of the disease or trait in question in individuals homozygous for either allele and in heterozygotes. Classically, it was thought that disease-associated alleles of SNPs did not directly cause disease, and that they were simply genetically-linked to (in linkage disequilibrium, LD, with) the true causative variant, typically at or controlling a protein-coding gene in the vicinity that was causing the disease. However, now that we know that 82% of the genome is functional [2], that dogma is being reassessed: these simple variants in the genome may directly functionally contribute to the risk of disease, rather than merely be in LD with a coding functional variant elsewhere. The emerging post-ENCODE model thus stipulates that disease-associated SNPs may be functional and contributing to the pathogenesis of the disease. Indeed, contrary to early expectations, the vast majority (~93%) of disease- and trait-associated SNPs emerging from GWAS lie within a non-coding sequence, which includes intergenic and intronic regions, promoter regions, small ncRNA as well as lncRNA genes, antisense transcriptional units, and enhancer or insulator regions, and therefore they are likely to influence gene regulation [12,13,14]. Accordingly, in parallel with increased insights into the non-coding genome, emerging studies are characterizing SNPs that are located within non-coding RNA regions and are associated with various complex diseases, including diabetes and cancer [14]. The prevalence of diabetes, especially type 2 diabetes (T2D), and cancer has increased significantly in recent years, with a huge impact on health worldwide [15]. Epidemiologic evidence indicates that T2D and cancer often coexist in the same patients, and many risk factors, such as obesity, a sedentary lifestyle, smoking, and ageing, are common for both diseases [16]. Moreover, it is commonly understood that, while both genetic and environmental (including nutritional) factors contribute to T2D, the additive effects of common disease-associated genetic variants—which differ in different populations and parts of the world—are central to the etiology of this disorder and its relationship to cancer; however, the functional basis of the biological link between these diseases still needs to be better understood and emphasized. Interestingly, the United Arab Emirates (UAE) population has one of the world’s highest prevalence rates of diabetes (https://diabetesatlas.org/, accessed on 5 January 2022), and the burden of cancer is ranked as the second leading cause of non-communicable diseases (NCD)-related mortality in the country [17]. With the aim to highlight the importance of non-coding variants and lncRNA genes as a causative factor in both T2D and cancer, in this work we survey the major published UAE T2D GWAS to search for, and identify, putative new lncRNA-associated genetic etiologies that are shared between T2D and cancer in the UAE population.

## 2. Methods

### 2.1. Information Sources and Search Strategy

For this analysis, the input datasets were obtained from public GWAS of type 2 diabetes (T2D). The systematic literature search in PubMed was performed using the terms, “Type 2 Diabetes” and “GWAS” and/or “Genetic Variations” and “UAE” and/or “United Arab Emirates.” The two most recent datasets [18,19] were included in the analysis. Manual annotation of all published significant T2D-associated genetic variants outside of protein-coding genes was conducted by using the public web-based bioinformatic tool, UCSC Genome Browser [20] (https://genome.ucsc.edu/, accessed on 10 March 2022). The location of the genetic variant (for example, in an exon or intron of a lncRNA gene), the epigenetic and expression profile of the region containing the variant, in thousands of human samples and tissue types surveyed across the datasets (Epigenome Roadmap, GTEX, FANTOM, mRNA/EST, others) in the Browser, the evolutionary conservation of the region, all ENCODE Consortium experimental data from all ENCODE data tracks in the Browser spanning the region and judged in our manual annotation as pertinent to its function, and all other data necessary to determine the relevance and potential function of the SNP were all analyzed in the UCSC Genome Browser. The SSTAR functionality of the FANTOM5 website https://fantom.gsc.riken.jp/5/sstar/Main_Page, accessed on 10 March 2022 [21] was also used to identify the top 10 tissue types, primary cell cultures, and/or cell lines where the gene/s associated with SNPs have the highest expression. For genes with multiple promoters in SSTAR, we have aggregated data across the promoters.

### 2.2. Eligibility Criteria

The tables referenced in our study were obtained from, and correspond to the main and supplementary tables of, the two major selected GWAS datasets [18,19]. The typical data set input table format, per line, included a reference SNP number (“rs” followed by a number) alongside an associated gene, a “mapped gene”, or a “reported gene.” An rs number is a universal SNP ID, that allows for searching of the SNP in any database. We manually analyzed all SNPs that were located in lncRNA genes (as evident from the gene name) or non-coding regions, and also all SNPs that were genomically complex. The latter was defined as one or more of the following: the presence of discrepancy between the mapped and reported gene names, or more than one gene (protein-coding and/or non) associated to the SNPs in the input data, or an alphanumeric gene name that was not corresponding to a protein family name/function, for example, gene names such as those that began with “AC” followed by numbers, or those displaying a mixture of letters and numbers without a gene-family root, and those that began with “Linc” (long intergenic non-coding RNA) and “Loc” (Locus, of presumably unknown function), which usually refer to lncRNA genes, and genes of unknown function many of which may be non-coding, respectively, were considered as indicators of SNPs potentially located in lncRNAs. All the SNPs in the dataset that met one or more of these criteria were then highlighted and annotated.

### 2.3. Data Collection and SNP Annotation

A total of 1284 SNPs were extracted from the GWAS input datasets. This number was then narrowed to 1132 by removing all the redundant SNPs. Next, based on the approach described in “Eligibility criteria” above, 370 SNPs were highlighted as potentially located in lncRNA genes (flow chart). Finally, the most promising SNPs (64 total) were thoroughly analyzed and annotated. The UCSC Genome Browser [20,22] was used to analyze and manually annotate every GWAS SNP. Upon selecting the GRCh37/hg19 human genome assembly, the UCSC Genome Browser was configured incorporating all pertinent browser tracks. The options “pack” and “show” for the selected tracks were chosen. From the default tracks that are displayed under the “Genes and Gene Predictions” Category, the “UCSC genes,” “NCBI Ref Seq” and “GENCODE” were chosen. For the “Phenotype and Literature” category, the “GWAS catalog” and “SNPedia” tracks were chosen. Further, for the “mRNA and EST” category, “Human ESTs” and “Human mRNAs” were selected whereas for the “Expression” category, the “Gtex GeneV8” was set to “full”. We used the ENCODE Regulation track of the UCSC Genome Browser to examine the H3K4Me1 and H3K27Ac (enhancer) signatures, and we viewed the DNAse I cluster track, to determine whether the SNPs were located in areas with chromatin signatures consistent with enhancers in ENCODE Tier 1 cell types and in open-chromatin regions based on DNAseI-seq of 125 ENCODE cell types, respectively. All other settings that are not mentioned were set to “hide”. To find the SNPs in the genome, the SNP ID was copied into the search box of the UCSC Genome Browser. Then all the SNPs located in non-coding regions were annotated as well as assessed for relevance to cancer or T2D. The relationship to cancer was annotated both in the case of a direct correlation to a specific cancer, and in non-cancer- specific contexts but still cancer-related, including any mention in the associated literature of phenomena such as disabling of a tumor suppressor gene, apoptotic factors, and mitotic activity regulation. Similarly, we qualitatively assessed the relationship to diabetes. Lastly, when the highlighted SNP was surrounded by a large number of other SNPs that are also significantly associated with the same disease or trait or with other diseases or conditions in GWAS data in the NHGRI/EBI track of the Browser (a phenomenon we term a “SNP cloud”), as shown in Figure 1, the publications and gene names linked to all those nearby SNPs were analyzed for verifying any potential relevance to both T2D and cancer.

## 3. Results

### 3.1. Customized Annotation Pipeline SNPs Selection

In this study an in-depth re-annotation of GWAS studies [18,19], in order to identify all statistically significant genetic variants residing in non-coding regions of the genome or in lncRNA genes, and that are of joint relevance to T2D and cancer in the UAE population was performed. The input dataset was obtained from public GWAS of T2D in the UAE population. From the few studies performed in this region and field, the two most recently published datasets in the UAE population [18,19], as these were applicable to our project, were selected. A total of 1284 SNPs associated with T2D and obesity were extracted from the main and supplementary tables of the GWAS input datasets [18,19]. The total number was then reduced to 1132 SNPs because we identified and excluded 152 redundant SNPs within and between the input datasets. This number was further narrowed to 370 by removing all the SNPs (762) associated with protein-coding genes. Finally, 67 SNPs of the 370 marked as potentially located in lncRNA genes or in non-coding regions (Figure 2), were selected for further analysis. These SNPs were confirmed (except for those otherwise mentioned below) for residing in non-coding regions. It is widely known that genetic variants are prone to mis-annotation and there is a substantial bias toward protein-coding genes in SNP annotations and in the related literature [23,24]; therefore, an unbiased reannotation was essential, and we used the UCSC Genome Browser to implement such a reannotation. In this regard, out of the 64 selected SNPs, 13 were classified as “corrections” because of discrepancies between the previous annotation and our interpretation of the UCSC Browser results. These discrepancies were mainly related to the name of the nearest SNP-associated gene and/or the actual location in the genome. Furthermore, five SNPs, previously annotated in non-coding regions of the genome, were instead confirmed as located in protein-coding regions; regardless, we proceeded to annotate these SNPs for any relevance to T2D and cancer. The remaining 59 SNPs were confirmed as residing in non-coding regions (Figure 2). Finally, six SNPs jointly associated with both cancer and T2D (rs1495741, rs1061810, rs2521501, rs8042680, rs7526425, rs2157719), implicating a genetic link between the two diseases, were recognized. Summary of the six SNPs is showed in Table 1 and in Figure 3 which illustrates the six SNPs, the chromosomal region where the SNPs are located and the nearest gene.

### 3.2. Characteristics of SNPs Associated with T2D and Cancer

#### 3.2.1. rs1495741

This SNP was previously reported as associated with the NAT2 (N-acetyltransferase 2) and PSD3 (Pleckstrin and Sec7 Domain Containing 3) genes. According to the NCBI Ref Seq track in the UCSC Genome Browser, NAT2 encodes an enzyme that functions to both activate and deactivate arylamine and hydrazine drugs and carcinogens [41,42] (Table 1 and Appendix A). Polymorphisms in this gene are also associated with higher incidences of cancer (e.g., lung cancer, esophageal squamous cell carcinoma, acute myeloid leukemia, and breast cancer) and drug toxicity [26,27,28,29]. Moreover, a recent study combining a GWAS meta-analysis of 2764 individuals with direct, reference measures of insulin sensitivity with functional validation both in vitro and in vivo, identified NAT2 as a novel insulin sensitivity locus [43]. A second arylamine N-acetyltransferase gene (NAT1) is located near NAT2 (Appendix A). PSD3 is a protein-coding gene associated with hepatocellular carcinoma (HCC), one of the most common types of primary liver cancer that often occurs in people with chronic liver diseases [25]. Risk factors generally include those which cause chronic liver disease, such as viral hepatitis B and C [44] but also metabolic disorders such as T2D and obesity [45]. The presence of two associated genes suggested that this SNP could reside in a genomically complex locus [46]; complex loci often, though not always, contain lncRNA genes. In this case, our analysis confirmed rs1495741 as an intergenic variant located between PSD3 and NAT2, at approximately 14 Kb from the 3′ end of the NAT2, and approximately 100 Kb from the 3′ end of PSD3. NAT2 is thus the nearest gene to the SNP (Table 1 and Figure 3A). The genomic region within boundaries of NAT2 and PSD3 genes is well-conserved across primates, with the highest level of conservation in the chimpanzee, as expected. Partial conservation of this region is observed in pig, mouse and rat; however, the genomic region around the SNP (100 bp) is not conserved in gorilla, mouse and rat, whereas in the pig it is only partially conserved. We located NAT2 in the FANTOM5 SSTAR database and reviewed both of the annotated transcription start sites (TSSs, promoters). The highest expression was in liver and hepatocytes from different donors, small intestine, colon, and fetal duodenum. The GTEx RNA-seq track of the UCSC Browser, which shows median gene expression levels in 52 tissues and 2 cell lines, indicates that NAT2 has the highest median expression in liver, whereas PSD3 has the highest median expression in the brain. For rs1495741, our review of the ENCODE Regulation track of the UCSC Genome Browser showed a lack of DNAse I hypersensitive site signatures overlapping the SNP. The H3K4Me1 and H3K27Ac ChIP-seq signals were also minimal and arose from different cell types, suggesting a lack of enhancer signatures overlapping this SNP. No evidence of any transcription factor binding sites overlapping this SNP was observed in ENCODE ChIP-Seq data for 161 transcription factors. The only transcription factor binding sites near this SNP was GATA3. Regarding SNP-disease association as evidenced by the NHGRI-EBI GWAS Catalog track of the UCSC Genome Browser, rs1495741 is reported in 14 published papers where it is mainly associated with triglycerides, cholesterol levels and metabolic disorders. Moreover, two GWAS studies associate this variant with bladder cancer risk. Therefore, summarily, while this is not a non-coding RNA variant, it is a non-coding (intergenic) variant at a complex locus clearly associated with diabetes in the UAE population as well as with cancer.

#### 3.2.2. rs1061810

The variant, rs1061810, was previously reported associated with two GENCODE Transcript annotations, ENST00000530450.1_3 (AC087521.2), and ENST00000637427.1_3 (AC087521.4) described as lncRNA and the protein-coding HSD17B12 (hydroxysteroid 17-beta dehydrogenase 12). Our analysis found this SNP exonic to the protein-coding gene HSD17B12, specifically it falls in the 3′ UTR (Figure 3B); whereas, relative to the two non-coding genes at this locus, rs1061810 is intronic to AC087521.2, and exonic to AC087521.4, which is antisense to HSD17B12 (Table 1, Figure 3B and Appendix A). Based on the NCBI Ref Seq track in the UCSC Genome Browser, HSD17B12 encodes for the enzyme 17 beta-hydroxysteroid dehydrogenase (17beta-HSD) that converts estrone into estradiol (E2) in ovarian tissue, but it is also involved in fatty acid elongation. The fatty Acyl-CoA biosynthesis and metabolism of steroid hormone pathways are both related to this gene and they are both linked with cancer. In fact, estrogen is well-known as a proliferative hormone and a driver of oncogenic and proliferative gene networks in estrogen receptor positive breast cancer, where it serves as a nuclear hormone; indeed, upon binding its receptor, it leads to the internalization of the receptor which in turn serves as a transcription factor, transitioning to the nucleus where it binds the promoters of the oncogenes that it activates (and of the tumor suppressors that it represses) [47,48]. Fatty acids are key players in cellular processes (cellular bioenergetics, membrane biosynthesis and intracellular signaling) involved in cancer development and progression [49]. In this regard, a study in human breast carcinoma suggested that involvement of HSD17B12 in the growth of carcinoma cells is not necessarily linked to the peripheral E2 biosynthesis but rather to the synthesis of very long chain fatty acids (VLCFAs), such as arachidonic acid, which contributes to breast carcinoma progression [30]. Furthermore, HSD17B12 is a marker of poor prognosis in ovarian carcinoma [31] and it is also associated with cutaneous melanoma [32] (Table 1 and Appendix A). Rs1061810 is associated with T2D in two GWAS studies in the European population [50,51], highlighting the importance of this locus in T2D susceptibility in diverse populations and not only in the Middle East. The conservation of the locus where the variant rs1061810 resides is high across primates, and partial in the mouse, pig and dog. The conservation level is low in rat. We located HSD17B12 in the FANTOM5 SSTAR database and reviewed all five TSSs (FANTOM5-annotated clustered promoters). The top ten cell lines where HSD17B12 is mostly expressed are: mast cells, the gall bladder carcinoma cells, smooth muscle cells from the aorta, mesenchymal stem cells, macrophages, adipocytes from two different donors, melanocytes, fibroblasts from aortic adventitial and fibroblasts from skin. For rs1061810, our review of the ENCODE Regulation track of the UCSC Genome Browser showed a lack of DNAse I hypersensitive site signatures overlapping the SNP. A strong signal was observed in the RNA-seq subtrack in all nine ENCODE Tier 1 cell lines. The signal was weak in the H3K4Me1 and H3K27Ac ENCODE histone modification ChIP-seq data for ENCODE Tier 1 cell types. No evidence of any transcription factor binding sites overlapping or near this SNP was observed in ENCODE ChIP-Seq data for 161 transcription factors.

#### 3.2.3. rs2521501

Previously mapped to the FURIN-FES locus, rs2521501 is intronic to FES (Figure 3C), whereas FURIN is the second nearest gene (approximately 10 kb) to this SNP (Table 1 and Appendix A). According to the NCBI Ref Seq track in the UCSC Genome Browser, the proto-oncogene FES (Feline Sarcoma) encodes the human cellular counterpart of a feline sarcoma retrovirus protein with transforming capabilities. The gene product has tyrosine-specific protein kinase activity, which is required for the maintenance of cellular transformation. Its chromosomal location is linked it to a specific translocation event identified in patients with acute promyelocytic leukemia [33]; sarcoma is another type of cancer associated with FES [34]. The FURIN gene encodes for a calcium-dependent serine endoprotease which is expressed in many tissues, and it is involved in various physiological and pathophysiological processes ranging from embryonic development to carcinogenesis [52]. This endoprotease has rocketed to prominence in the past two years as a consequence of its role as a SARS-CoV-2/COVID-19 co-factor (which cleaves the S-protein) [53]. Several reports suggested that FURIN inhibition can suppress the tumorigenic properties of various cancer cell types, while other studies reported instead that FURIN inhibition may lead to a more aggressive phenotype of cancer cells [54]; however, despite these controversies, it is well established that FURIN plays a key role in cancer [54]. The intronic region of the FES gene where the SNP is located (approximately 1400 bp) is well conserved only in some primates, such as the chimpanzee, gorilla, rhesus monkey, crab-eating macaque, and green monkey. In rat, mouse and pig the conservation level is very low with a partial conservation only in the region around the SNP (70 bp). We located FES in the FANTOM5 SSTAR database and reviewed all four TSSs (FANTOM5-annotated clustered promoters). The highest expression was in different cell lines of CD14+ monocytes (from different donors), CD14+CD16− monocytes (different cell lines from different donors), biphenotypic B myelomonocytic leukemia cells, acute myeloid leukemia (FAB M5) cells, and eosinophils (different cell lines from different donors). The GTEx track indicates FURIN has the highest median expression in the liver, followed by the pancreas (organs directly relevant to T2D and obesity pathogenesis), whereas FES has the highest median expression in the spleen followed by the lungs. In several GWAS studies, as evidenced by the NHGRI-EBI GWAS Catalog track of the UCSC Genome Browser, rs2521501 is associated with diastolic and systolic blood pressure as well as with the interaction of blood pressure with alcohol consumption and/or cigarette smoking. The ENCODE Regulation track of the UCSC Genome Browser shows a lack of DNAse I hypersensitive site signatures overlapping the SNP. A low level of expression was observed in the RNA-seq data from the ENCODE Tier 1 nine cell lines, consistent with the intronic localization of the SNP, as the signal was greater in the gene’s exons as expected. The signal was also weak in the H3K4Me1 and H3K27Ac ENCODE histone modification ChIP-seq data for ENCODE Tier 1 cell types. For rs2521501, there was no evidence of any transcription factor binding sites overlapping, or near this SNP in ENCODE ChIP-Seq data for 161 transcription factors.

#### 3.2.4. rs8042680

PRC1 and PRC1-AS1 were previously reported as the closest genes to rs8042680; we found that this SNP is intronic to both genes (Table 1, Figure 3D and Appendix A). PRC1 (protein regulator of cytokinesis 1) encodes for a protein involved in cytokinesis and microtubules organization. PRC1 is overexpressed in human hepatocellular carcinoma cells and it is associated with the increased chemoresistance of these cells [55]. PRC1-AS1 is an antisense lncRNA and, similarly to PRC1, is associated with hepatocellular carcinoma [35]. The intronic region where the SNP is located is conserved only in primates with the exception of marmoset and squirrel monkey, while in rat, mouse and pig it is not conserved. We were unable to find PRC1-AS1 antisense lncRNA in the FANTOM5 SSTAR database, and therefore visually reviewed that lncRNA gene in FANTOM CAGE data via the graphical browser (testis, tongue, mesothelioma, chondrocyte, and mesenchymal stem cells were the top expressors). We located PRC1 in the SSTAR database and reviewed all four clustered TSSs that have precomputed FANTOM5 expression data. The highest expression was in reticulocytes, hepatoblastoma, osteosarcoma, bone marrow, and CD14+ monocytes (the latter were in the top 15 expressors for the antisense too). With regards to tissue expression, PRC1 has the highest median expression in fibroblasts and EBV-transformed lymphoblastoid cell lines, while PRC1-AS1 has the highest median expression in testis. For rs8042680, our review of the ENCODE Regulation track of the UCSC Genome Browser shows a lack of DNAse I hypersensitive site signatures overlapping the SNP. The H3K4Me1 and H3K27Ac ChIP-seq signals were also minimal and arose from different cell types, suggesting a lack of enhancer signatures overlapping this SNP. No evidence of any transcription factor binding sites overlapping or near this SNP was observed in ENCODE ChIP-Seq data for 161 transcription factors. As evidenced by the NHGRI-EBI GWAS Catalog track of the UCSC Genome Browser, two papers reported the association of rs8042680 with T2D. Summarily, hence, this is another SNP that is associated both with T2D in the UAE population and with cancer.

#### 3.2.5. rs7526425

From the input datasets’ published annotations, rs7526425 was reported as mapped near three genes: AC105275.1, SLC30A1, and RD3. The SNP is not found in the “NHGRI-EBI Catalogue of Published GWAS” track of the UCSC Genome Browser (Table 1 and Appendix A). RD3 is the closest gene to the SNP (Figure 3E); this gene encodes a retinal protein that is associated with promyelocytic leukemia-gene product (PML) bodies in the nucleus. Moreover, RD3 plays a regulatory role in neuroblastoma progression and its loss is associated with aggressive neuroblastoma and poor clinical outcomes [36]. The gene AC105275.1 encodes a lncRNA and is antisense to RD3. The gene SLC30A1 encodes a zinc transporter protein, which is involved in maintaining cellular zinc homeostasis in mammalian cells. This gene is therefore relevant to diabetes, since zinc is an essential co-factor for insulin metabolism in the pancreatic β-cell [56]. Moving on to the tissue expression of the genes, RD3 has the highest median expression in the pituitary and SLC30AL has the highest median expression in the liver; both organs are directly relevant to diabetes pathogenesis and etiology. The genomic region around the SNP (100 bp) is highly conserved across most primates, as expected and is well-conserved in rat, mouse, rabbit, pig and dog. We found RD3 in the FANTOM5 SSTAR database and reviewed all three TSSs (FANTOM5-annotated clustered promoters). The highest expression was in the pineal gland, retinoblastoma cell line Y79, fetal eye, lung carcinoma cells, and medulloblastoma cells. Finally, the UCSC Browser shows a large SNP cloud surrounding the SNP. This is hence yet another SNP with a dual diabetes and cancer association, and is a UAE T2D susceptibility risk variant per the input datasets. A compelling case can be made for a non-coding regulatory region encompassing this SNP (Appendix A) as being even more relevant than RD3 as an explanation for the SNP’s functional significance: the SNP falls directly into a prominent ENCODE DNAse I hypersensitive region that also corresponds to a transcription factor binding (ChIP-seq) consensus signature across 13 of the 125 ENCODE cell types directly overlapping the SNP. In addition, there was a strong H3K4Me1 signal and weak H3K27Ac signal in the ENCODE histone modification ChIP-seq data for ENCODE Tier 1 cell types, also directly overlapping the SNP. The two enhancer histone modifications occurred only in GM12878 cells, which were also one of the 13 DNAse I hypersensitive cell types in the region. Summarily, there is evidence for open chromatin and putative enhancer modifications directly encompassing this SNP in GM12878. For rs7526425, there are three ENCODE TFBS ChIP-seq hits directly overlapping and encompassing the SNP: RELA, POU2F2, and MEF2A. While all three contain consensus recognition sites for the respective TFs, none of the consensus sequences overlaps the SNP (whereas the rest of the ChIP-seq-detected binding site contains the SNP). Therefore, the SNP is in TFBSs, but is unlikely to affect binding affinity, due to its location outside of the consensus. Interestingly, all three binding signals include GM12878 cells, where the DNAse I and enhancer signatures also occurred. The regulatory significance of the region containing this variant in GM12878 is also supported by the multiple Hi-C interactions found ~130 bp away with multiple nearby genomic regions in the UCSC Genome Browser’s Rao et al. 2014 Hi-C track.

#### 3.2.6. rs2157719

As previously reported, rs2157719 is within the CDKN2B-CDKN2A gene cluster. The closest protein-coding gene to this SNP is CDKN2B (cyclin dependent kinase inhibitor 2B) (Figure 3F) which is adjacent to the tumor suppressor gene CDKN2A (Cyclin-Dependent Kinase 4 Inhibitor A) in a region that is frequently mutated and deleted in a wide variety of tumors (Table 1 and Figure 2). Both genes are associated with several cancer types, including ovarian cancer, pancreatic cancer, melanoma and, head and neck squamous cell carcinoma [37,38,39,40]. Rs2157719 is also intronic to the non-coding RNA, CDKN2B antisense RNA 1 (CDKN2B-AS1), also known as ANRIL (Figure 3E), which is linked to the progression of diabetic nephropathy, one of the most common T2D complications [57]; however, the prominence of CDKN2B-AS1 in the literature to date is mostly due to its role in cancer, where it can enhance cell proliferation, cell cycle progression, and inhibit apoptosis and senescence. Moreover, CDKN2B-AS1 is overexpressed in many cancer types and is a well-recognized prognostic and diagnostic biomarker in cancer [58,59,60]. A close examination of the “SNP cloud” surrounding rs2157719 (Figure 1) reveals that several SNPs in this region are significantly associated with other cancer risks; while consistent with the linkage disequilibrium expectation that a genetic region containing multiple SNPs is associated with the risk, this is surprising because of the diversity of these cancers. For instance, rs1333048 (Figure 1) is a biomarker of breast cancer susceptibility, and it is associated with the risk of toxicity after chemotherapeutic drug (cisplatin) treatment in lung cancer patients [61,62]. Moreover, rs4977574 is associated with kidney cancer development in the Ukrainian population [63] while rs10757278 is associated with ANRIL expression levels and cisplatin resistance in cancer [58,61] (Figure 1). The GTEX track of the UCSC Genome Browser shows that CDKN2B-AS1 has the highest median expression in the colon and small intestine. From GWAS studies reflected in the NHGRI-EBI track of the UCSC Browser, rs2157719 itself is associated with diverse cancers; the SNP has been referenced in several papers as related to glaucoma and glioma [64,65,66,67]. The intronic region where the SNP is located is highly conserved across most primates, as expected. Moreover, the region is partially conserved in rat, mouse, rabbit, pig and dog. We located both CDKN2B and CDKN2B-AS antisense lncRNA in the FANTOM5 SSTAR database. For CDKN2B we reviewed all two TSSs (FANTOM5-annotated clustered promoters) and found that the highest expression for this gene was in lens epithelial cells, preadipocytes, mesenchymal cells, cardiac fibroblasts, adipocytes, differentiated osteoblast, and melanoma cells. Meanwhile, for the CDKN2B-AS antisense lncRNA, we reviewed all three promoters and found the highest expression in carcinosarcoma cells, gastrointestinal carcinoma cells, osteosarcoma cells, lens epithelial cells and gall bladder carcinoma cells. For rs2157719, there was a DNAse I hypersensitive region in 8 of the 125 ENCODE cell types directly overlapping the SNP. In addition, there was a strong signal in the transcription track showing the transcription levels assayed by RNA-seq on nine cell lines, with the highest signal for the HeLa-S3 cells. There was a modest H3K4Me1 signal (highest in the Epidermal Keratinocyte Cells NHEK), and weak H3K27Ac signal in the ENCODE histone modification ChIP-seq data for ENCODE Tier 1 cell types also directly overlapping the SNP. Summarily, there is evidence for open chromatin and putative enhancer modifications directly encompassing this SNP in HeLa-S3 and NHEK cells. No evidence of any transcription factor binding sites overlapping or near this SNP was observed in ENCODE ChIP-Seq data for 161 transcription factors.

## 4. Discussion

Traditionally, proteins and small molecules, and the molecular pathways and regulatory networks containing them, have given rise to the majority of drug targets. For this reason, GWAS projects, despite being an inherently unbiased approach, have been historically biased toward the identification of protein-coding regions and genes carrying disease risks. Meanwhile, data on non-coding SNPs has either not been pursued or has been consistently misinterpreted in favor of distant protein-coding variants, while ignoring the nearby or overlapping lncRNA genes or non-coding regulatory regions that can be more likely causal determinants of the disease phenotype. Simply because 98.5% of the human genome is non-coding, and also because two-thirds of the approximately 60,000 human genes are lncRNA genes, most significant disease-associated variants are in non-protein-coding regions of the genome. Therefore, most GWAS studies that have limited their analysis to previously known, hence primarily protein-coding risk variants, despite having genotyped patients and controls on a whole-genome level, contain frequent and abundant mis-annotations, and encompass numerous lncRNA genes and non-coding regions in raw data, which are often subsequently not interpreted [68,69]. An unbiased reannotation of disease-associated SNPs is therefore essential and moreover, re-annotation from previous research findings can potentially result in a different knowledge outcome [70]. For example, in our work, we have uncovered previously-unknown dual diabetes–cancer associations of specific non-coding variants based on GWAS signals specifically from the UAE population. In this study, we performed an integrated re-annotation of recent UAE population specific T2D datasets, we highlighted numerous SNPs in non-coding regions and several in LncRNA genes, and we computationally defined the relevance to cancer of a subset of those regions. Previous studies, mainly observational epidemiological studies, have established associations between T2D and cancer in terms of shared common risk factors such as, hyperglycemia, hyperinsulinemia, obesity, a lack of physical activity and diet, but the biological mechanisms supporting these associations have remained poorly known [71]. Similarly, genetics studies, such as those based on the identification of the same genetic variant and risk allele that independently predisposes to both type 2 diabetes and cancer, are still insufficient to explain the increased risk of cancer in people with T2D [71]. Here, we addressed the knowledge gap in functionally and genetically understanding the diabetes–cancer connection: we successfully discovered six significant disease-associated genetic variants that signify new biological links between T2D and cancer. All six SNPs reside in non-coding regions: specifically, two SNPs (rs1495741, and rs7526425) are intergenic, three SNPs (rs1061810, rs8042680, and rs2157719) reside in lncRNA genes, and one SNP (rs2521501) is intronic to the FES gene. All six SNPs show a clear and strong association with both T2D and cancer; this is demonstrated not only by the GWAS studies reported in the NHGRI-EBI GWAS Catalog track, but also by the direct involvement, in both diseases, of the nearby, or overlapping, SNPs-associated genes. For instance, the variant rs1495741 is directly correlated with bladder cancer [72], elevated triglyceride levels, T2D risk [73], and liver injury [74]. Additionally, both the proximal and the distal gene to the SNP, NAT2 and PSD3, respectively, are strongly correlated with both cancer and T2D [26,43,75,76,77,78]. Another SNP significantly associated with T2D and cancer is rs2157719; this variant is indeed directly correlated to glaucoma and glioma [64,65,66,67] and resides in a genomic region that is frequently mutated and deleted in a wide variety of tumors. Moreover, it is intronic to the lncRNA CDKN2B-AS1, which is aberrantly expressed in various malignancies and it is also implicated in numerous non-malignant diseases, including diabetes and its complications [79]. Several SNPs are associated with genes that play a key role in T2D and, according to the GTEx RNA-seq track of the UCSC Genome Browser, have the highest median expression in the liver; this is the case of NAT2 (rs1495741), FURIN (rs2521501), and SLC30A1 (rs2157719). NAT2 is a cellular enzyme involved in the metabolism of a variety of different compounds, including carcinogens. Deficiency of this gene causes mitochondrial dysfunction with decreased cellular respiration and ATP generation, suggesting that NAT2 may mediate insulin resistance and mitochondrial dysfunction by binding key regulators of energy balance [80]. FURIN is a membrane-bound protease broadly involved in the maintenance of cellular homeostasis. A recent study showed that individuals with high plasma furin levels concentration have an elevated risk of diabetes mellitus and premature mortality [81]. Moreover, furin protein upregulation results in worse outcomes in diabetic patients with SARS-CoV-2 infection [82]. Finally, SLC30A1 plays a key role in maintaining the cellular zinc levels within a physiological range [83]. Zinc is a second messenger that controls many processes associated with insulin signaling [84,85]. Furthermore, all these genes are mostly expressed in the liver, an organ that plays a key role in the regulation of glucose metabolism [86]. T2D is an excellent setting for RNA therapeutics because much of the pathogenesis occurs in the liver, and RNA drugs injected into the bloodstream naturally go to the liver but not to any other organs [87]. RNA therapeutics target mRNA as well as non-coding RNAs, including lncRNAs, with small interfering RNAs (siRNA), antisense oligonucleotides (ASO), modified-backbone oligonucleotides (MBO), and other RNAi-based drugs [88]. RNA drugs are highly sequence-specific, as each drug has only one target, can distinguish the risk and the non-risk alleles and hence can be used in patients that have the causal disease-risk variant. Therefore, RNA-based drugs have fewer side effects and are very stable. For these reasons, RNA drugs are destined to change the way disease is treated in a more targeted and personalized manner [87,88]. For example, the milestones achieved with the development and approval of Inclisiran as a synthetic siRNA drug against an RNA target, specific to the liver, illustrate the potential for the development of similar drugs targeting diabetes and cancer, deployed against other, including lncRNA, targets. Many diabetes genes are indeed expressed in the liver, which is easily targeted by this approach, and the GWAS-empowered discovery of non-coding, liver-expressed diabetes and cancer candidates can generate a target list for future development of Inclisiran-class drugs.

Limitations of our study include how the GWAS data had been analyzed by the teams that provided the input datasets, and the incomplete availability of biological data types as well as experimental datasets from specific cell and tissue types in the UCSC Genome Browser. We focused on published GWAS datasets of SNPs deemed jointly relevant to T2D and cancer, analyzing these variants’ relevance to the non-coding regions of the genome, specifically lncRNA; however, GWAS approaches are inherently prone to false positives and false negatives with regards to the association of a variant with a phenotype: there is a possibility of SNPs in the datasets that are unrelated to the diseases (false positive) or SNPs that are related that were not included at all (false negative). This could be due to ascertainment biases, weak association in a common disease/common variant context where a single SNP fails to reach significance, a population-specific lack of genotype–phenotype association that exists in other population, a smaller study population and, thus, sample sizes utilized by the authors of these published papers on GWAS, or other factors. Additionally, most GWAS studies, including those that performed high-throughput genotyping rather than de-novo sequencing of multiple individual genomes, are based on previously known variants, resulting in the exclusion of newer variants that may have a stronger association or may be located nearer to the actual functional feature in the genome. The impact of this limitation leads to the exclusion of relevant and possibly targetable SNPs (false negative) and the inclusion of SNPs that are unrelated (false positives) to T2D. The UCSC Genome Browser may not contain expression data for certain low-abundance transcripts that lack cDNA/EST and ENCODE RNA-seq representation. In addition, the UCSC Genome Browser lacks representation of certain datasets that could be essential to our study and that have been published but are not reflected in the Browser’s tracks. While our work should facilitate the development of targeted therapies, a limitation is that the identified genes would need to be the direct functional causes of the phenotype (pending functional validation experiments) in order to be targeted; the specific risk alleles of the SNPs should be targeted in a sequence-specific fashion if causative; and the expression of the target would have to be confined to tissues and organs where it directly underlines the disease phenotype, so as to avoid off target effects when sequence-based, such as RNAi, therapies are deployed. Despite possibly being limited to the specific populations of the UAE, this work provides genetics-based evidence of novel non-coding biological links between T2D and cancer, which feature specific putative regulatory regions and lncRNA genes. This work lays the foundation to further explore the shared genetic architecture between these two common diseases, as well as to expand the study to multiple populations [89]. Functional studies will be needed in order to understand the functional implications of these SNPs, and to provide new insights about the mechanisms leading to the progression of the two diseases.

## 5. Conclusions

In conclusion, we have gathered genetic-based evidential proof of a biological link between T2D and cancer. Our major findings include the successful discovery of six significant disease-associated genetic variants, identified in the UAE population, that have established a previously unknown biological link between T2D and cancer. The novelty of our study is in leveraging upon UAE-origin GWAS T2D data (given the relative lack of metabolic-disease GWAS in the Gulf and Middle East region) to identify variants of joint relevance in cancer and diabetes. These variants suggest specific functional links between those two common diseases, links that help to better explain their casually-known co-occurrence. These variants resided in lncRNA genes and at or near protein-coding genes. Genetic heterogeneity is central in common diseases (such as diabetes and cancers) that have a partly-genetic etiology; therefore, it is crucial to understand the genetic basis of a disease in a specific population, in addition to considering risk variants common to multiple populations. They provide an early foundation for a shortlist of possible drug targets, including those personalized toward variants occurring in the Gulf and Middle East/North Africa regions but not globally.

## Figures and Tables

**Figure 1 cancers-14-03313-f001:**
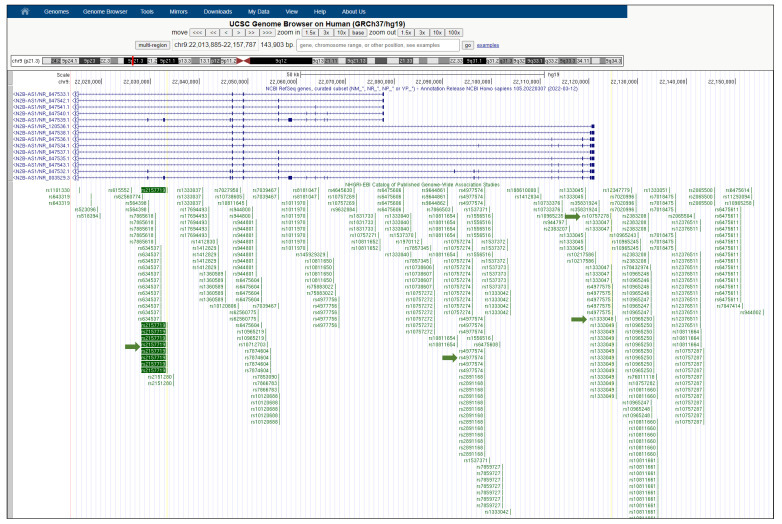
UCSC Genome Browser view of the human CDKN2B-AS1 gene. CDKN2B-AS1 (CDKN2B antisense RNA 1) is a lncRNA gene, also known as ANRIL. The SNP rs2157719 is intronic to CDKN2B-AS1 and it is surrounded by a large number of other significantly disease-associated SNPs in close proximity on the NHGRI-EBI GWAS Catalog track of the UCSC Genome Browser (a “SNP cloud”). Green arrows indicate the SNP rs2157719 associated with T2D and cancer, and the SNPs rs1333048, rs4977574, and rs10757278 associated with cancer.

**Figure 2 cancers-14-03313-f002:**
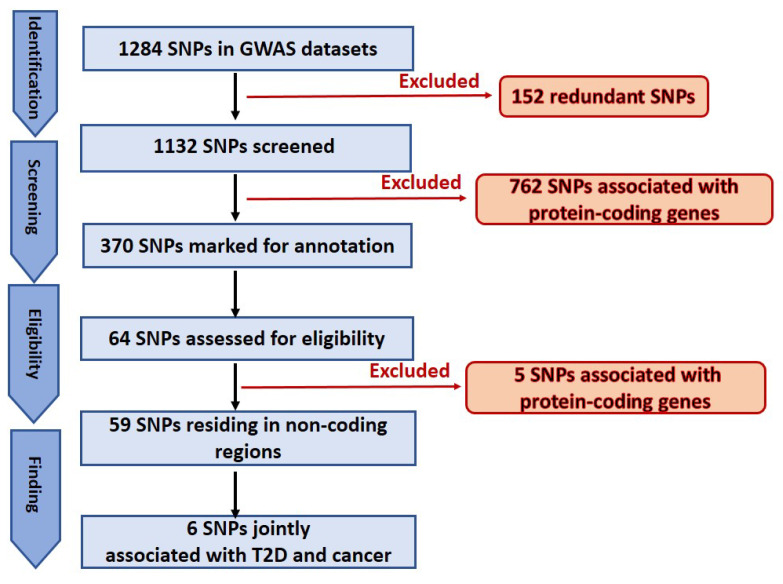
Schematic representation of SNPs selection.

**Figure 3 cancers-14-03313-f003:**
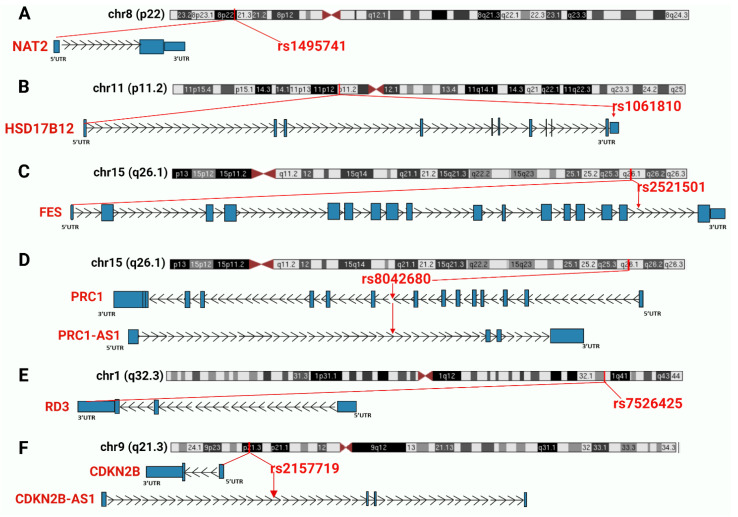
Summary of the chromosomal regions and the nearest genes of the six SNPs jointly associated with T2D and cancer. SNPs and nearest genes are marked in red. Introns are represented as lines with arrows indicating the direction of transcription, while coding exons are represented by blocks. (**A**) NAT2 gene and rs1495741; (**B**) HSD17B12 gene and rs1061810; (**C**): FES gene and rs2521501; (**D**) PRC1 gene, PRC1-AS1 antisense lncRNA and rs8042680; (**E**) RD3 gene and rs7526425; (**F**): CDKN2B gene, CDKN2B-AS1 antisense lncRNA and rs2157719.

**Table 1 cancers-14-03313-t001:** Summary of the six SNPs jointly associated with T2D and cancer.

SNP ID	Chromosomal Region	Nearest Gene	Nearby Genes	Type of Cancer	References
rs1495741	8p22	NAT2	NAT2, PSD3	HCC, LC, ESCC, AML, BC	[25,26,27,28,29]
rs1061810	11p11.2	HSD17B12	HSD17B12, AC087521.2, AC087521.4	BC, OC, MM	[30,31,32]
rs2521501	15q26.1	FES	FURIN, FES	APL, SARC	[33,34]
rs8042680	15q26.1	PRC1, PRC1-AS1	PRC1, PRC1-AS1	HCC	[35]
rs7526425	1q32.3	RD3	AC105275.1, SLC30A1, RD3	APL, NB	[36]
rs2157719	9p21.3	CDKN2B	CDKN2A, CDKN2B	OC, PC, MM, HNSCC	[37,38,39,40]

HCC: HepatoCellular Carcinoma; LC: Lung Cancer; ESCC: Esophageal Squamous-Cell Carcinoma; AML: Acute Myeloid Leukemia; BC: Breast Cancer; OC: Ovarian Cancer; MM: Malignant Melanoma; APL: Acute Promyelocytic Leukemia; SARC: Sarcoma; NB: Neuroblastoma; PC: Pancreatic Cancer; HNSCC: Head and Neck Squamous Cell Carcinoma.

## Data Availability

The data presented in this study are available on request from the corresponding author.

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
