# Peer review of "LncRNA-Associated Genetic Etiologies Are Shared between Type 2 Diabetes and Cancers in the UAE Population"

_cancers, 2022, doi:10.3390/cancers14143313_

Round 1
Reviewer 1 Report
In the manuscript entitled with “LncRNA-associated genetic etiologies are shared between type 2 diabetes and cancers in the UAE population”, Giordo et al annotated 6 SNPs significantly associated with both T2D and cancers, based on public dataset. Interestingly, the nearest gene of each of these 6 SNPs has important biological functions. This topic is quite interesting; however, this study is quite preliminary and I do not find any direct association of these SNPs with adjacent genes.
My main comments are as follows:
- No functional link or annotation for these 6 SNPs. HiC data in T2D or cancer related samples, if available, might provide some clue regarding the interactions between SNPs and the nearest gene.
- The authors claim that Rs7526425 is located in the active regulatory region (high H3K27ac and transcription factors signal), does this SNP affect transcription factor binding or transcription activity? Luciferase reporter assay would provide more convincing evidence.
Reviewer 2 Report
Giordo et. al performed a very interesting research establishing a biological link between type 2 diabetes and cancer. The link between the two disease, although so far away clinically suspected, seems to be linked in lncRNA genes.
I really think that authors performed a very good study who can improve the knowledge of these two - unfortunately - very common disease.
Author Response
We thank the reviewer for the positive feedback.
Reviewer 3 Report
In the present study entitled “LncRNA-associated genetic etiologies are shared between type 2 diabetes and 2 cancers in the UAE population.” the authors have identified six SNPs, residing in lncRNA genes or near protein-coding genes, associated with both type 2 diabetes and cancer in the UAE population, by analyzing GWAS datasets.
This is a comprehensive and very interesting study. Overall, the manuscript merit publication, however, the authors are recommended to address the following minor issues in order to be able to publish their results.
- The authors should improve the quality of figure 1 and Suppl. figures
- The authors should add figure(s) illustrating all the significant SNPs that would include the specific genetic variants, the chromosomal region and nearest gene(s).
- The authors should include the limitations of their study and future perspectives in the discussion section.
Author Response
1. The authors should improve the quality of figure 1 and Suppl. figures
Response
We thank the reviewer for pointing that out. We have improved the quality of the suggested figures and replaced figure 1, and all the supplementary figures.
2. The authors should add figure(s) illustrating all the significant SNPs that would include the specific genetic variants, the chromosomal region and nearest gene(s).
Response
We appreciate the reviewer's suggestion. We have included Figure 3 which illustrate all the six significant SNPs, the chromosomal region where the SNP is located and the nearest gene.
3. The authors should include the limitations of their study and future perspectives in the discussion section.
Response
We thank the reviewer for pointing that out. We included the limitations of our study and future perspectives in the discussion section. Please find the new part in red inked (from line 523 to 555) in the revised version of the manuscript.
Reviewer 4 Report
The manuscript of Giordo et al. is about the identification of T2D and cancer-relevant SNPs within non-coding regions. They have used GWAS of T2D from the UAE population and filtered SNPs for presence of known cancer-related SNPs by literature searches.
This led to the identification of 6 SNPs associated with both cancer and T2D.
The manuscript is interesting and well-written.
Although the manuscript leaves a good impression, the manuscript could benefit from more bioinformatic analyses regarding the interesting SNPs. Therefore, this reviewer has the following major points:
Major:
1. A major aim of the 6 candidates in future would be to test their therapeutic relevance. Therefore, the authors should indicate the level of conservation of the ncRNAs or ncRNA-regions (locus conservation, transcript conservation; primates, mouse/rat, pigs).
2. What is the expression level of the genes, which might be affected by the SNPs? Therefore FANTOM5 data can be used to show the top10 cells/cell lines, where the genes are expressed highest.
3. Are these SNPs located in enhancer areas or dominant epigenetic marks?
4. Probably some of the SNPs could lie in important transcription factor motifs (although they are not in a promoter region). Any evidence for what?
Minor:
1. Abstract: please remove the word "extensively" since this hasn't been shown by multiple data.
2. Introduction: "that are devoid of Open Reading Frames". Please change to "without apparent protein-coding potential". Reason: Many lncRNAs have open reading frames that are not active or whose activity hasnt been tested. In fact you can find easily ORFs for many lncRNAs.
3. Introduction: line 55: "biological", please change to "biological and physiological".
4. Methods: Line 106: [DATE]. Please specify.
5. throughout the manuscript: Please remove the information like "organized in excel tables". This is obvious.
6. Results: line 173/174: Please include an Acc. No. or link to the GWAS UAE data used in this study.
7. Sup. Fig. 1: please remove the irrelevant information for PSD3. This would lead to an increase in the genomic locus visible and therefore makes it easier for the reader.
8. Although mentioned in the figures, please also add the genomic locations for the sup. figures in the legends, e.g. ChrE(hg19):XXX,XXX,XXX-XXX,XXX,XXX.
9. Methods: Line 146: Please change the dot in ""full."" to ""full"."
